# Sensing Characteristic Analysis of All-Dielectric Metasurfaces Based on Fano Resonance in Near-Infrared Regime

Yongpeng Zhao [1], Qingfubo Geng [1], Jian Liu [2,*] and Zhaoxin Geng [1,3,*]

1   School of Information Engineering, Minzu University of China, Beijing 100081, China;
    20302031@muc.edu.cn (Y.Z.); 23012784@muc.edu.cn (Q.G.)
2   Institute of Semiconductors, Chinese Academy of Sciences, Beijing 100083, China
3   Key Laboratory of Ethnic Language Intelligent Analysis and Security Governance of MOE, Minzu University
    of China, Beijing 100081, China
*   Correspondence: liujian@semi.ac.cn (J.L.); zxgeng@muc.edu.cn (Z.G.)

**Abstract:** A novel, all-dielectric metasurface, featuring a missing wedge-shaped nanodisk, is proposed to investigate optical characteristics. By introducing symmetry-breaking to induce Fano resonance, the metasurface achieves an impressive Q-factor of 1202 in the near-infrared spectrum, with a remarkably narrow full width at half maximum (*FWHM*) of less than 1 nm. The ability to adjust the wavelength resonance by manipulating the structure of the wedge-shaped nanodisk offers a simple and efficient approach for metasurface design. This breakthrough holds great potential for various applications in sensing and optical filtering, marking a significant advancement in the field of nanophotonics.

**Keywords:** metasurfaces; sensor; all-dielectric; near-infrared regime

## 1. Introduction

The plasmonic metasurfaces have unique advantages in manipulating transmitted and reflected electromagnetic wavefronts [1], which have great potential to play a role in the fields of surface-enhanced infrared absorption [2,3], holography [4,5], energy conversion [6], programmable metasurfaces [7,8], and optical vortices [9–11]. A new strategy for metasurfaces has been developed based on Fano resonance [12,13].

Fano resonance is a special physical phenomenon formed by the coupling between discrete and continuous states, with its resonant spectral lines exhibiting an asymmetric distribution [14–17]. In optics, Fano resonance can arise from the coupling between two resonant cavities, resulting in spectra with sharp and asymmetric resonant peaks near the resonance wavelength. By breaking the symmetry, the Fano resonance has been tuned at different wavelengths [18–21]. Metasurfaces, two-dimensional planes composed of subwavelength dielectric structures, offer the capability for arbitrary control over electromagnetic waves. Within metasurfaces, Fano resonance can be employed to achieve applications such as highly sensitive optical sensing, efficient optical switching, tunable optical nonlinearity, and slow light. For instance, the manipulation of Fano resonance can be realized in metasurfaces by introducing dielectric structures of different shapes, enabling the control and modulation of optical signals for applications such as sensing [22] and optical switches [23]. However, the intrinsic loss of metal still limits its wide application. Therefore, dielectric resonator antennas and dielectric nanoparticles have been introduced and investigated in the optical regime [24]. Dielectric nanoparticles could support magnetic and electric dipole Mie-type modes, thus exhibiting shallow intrinsic losses. Meanwhile, all-dielectric metasurfaces based on Fano resonance have been investigated. [25–29].

All-dielectric Fano resonance structures are mostly dependent on two or more near-field coupled dielectric structures within the unit, such as asymmetric double bars and multimer structures, which have stringent requirements for processing and fabrication,

and the coupling of adjacent units has the potential to affect the performance of the system. A single resonator could effectively avoid the coupling phenomenon of other units in the array, and it has a flexible geometric modulation space. For example, a single cubic resonator symmetry-breaking could induce Fano resonance based on GaAs with a Q-factor of up to 600 [30], which presented a method for designing dielectric metasurfaces. The Q-factor of the symmetry-breaking cubic resonator is lower than that of the traditional Fano resonance structures. Thus, a higher Q-factor single resonator based on the Fano resonance needs to be developed.

## 2. Design and Theoretical Analysis

An all-dielectric metasurface consisting of missing wedge-shaped nanodisks is proposed. The structure consists of an array of silicon nanodisks and $SiO_2$ substrate. The designed element is composed of a $SiO_2$ substrate and a missing wedge-shaped silicon nanodisk resonator array (Figure 1a). The entire structure exhibits periodic variation characteristics in the x- and y-directions. The top and cross-sectional views of the unit cell are shown in Figure 1b. The nanodisks are periodically arrayed with periods $Px = Py = 500$ nm. The radius ($R$) and slice angle ($\theta$) of the nanodisks are 200 nm and 160°, respectively. The thickness ($t$) of the missing wedge-shaped silicon nanodisk is 200 nm. The thickness ($H$) of the substrate is infinitely thick to $t$.

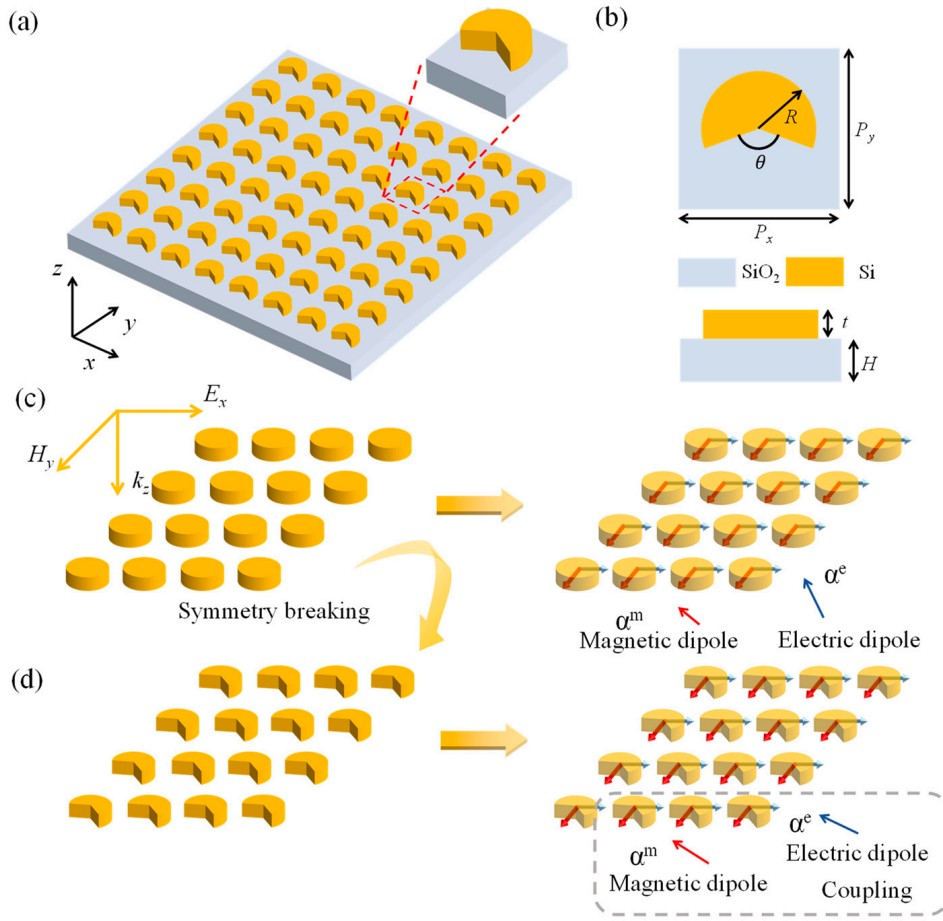

**Figure 1.** Schematic diagram of the all-dielectric metasurfaces and the working principle. (**a**) Array structure consists of the missing wedge-shaped nanodisk. (**b**) Top view and cross-section of a cell structure. (**c**) Top row shows the αe electric dipole mode and αm magnetic dipole mode of the incident x-polarized light excited nanodisk. (**d**) Bottom row shows symmetry-breaking and induced coupling of the αm magnetic dipole to the αe electric dipole.

To better understand the interaction between the dark mode and the bright mode of the electrical or magnetic resonance of an asymmetric all-dielectric metasurface, the electrical or magnetic resonance effects of a single all-dielectric hypersurface are first described. The principles underlying high-quality factor (Q) Fano metasurface are shown schematically in Figure 1c,d. The analytical expression for the reflection coefficient of a single nanodisk is as follows [31]:

$$r = \frac{2i \cdot \gamma_e \omega}{\omega_e^2 - \omega^2 - 2i\gamma_e\omega} - \frac{2i \cdot \gamma_m \omega}{\omega_m^2 - \omega^2 - 2i\gamma_m\omega} \tag{1}$$

where $\gamma_e$ is the electric dipole damping parameter; $\gamma_m$ is the magnetic dipole damping parameter; $\omega_e$ is the electric dipole resonance position; and $\omega_m$ is the magnetic dipole resonance position. The former term in the equation is associated with the electric dipole, and the latter with the magnetic dipole. The electric dipole has the same sign in both directions of the electric field component, whereas the magnetic dipole is different. This aspect could be used to distinguish between the known characteristics of electric and magnetic dipole radiation. Starting with a simple nanodisk atomic array discussion, there are $\alpha^e$ electric dipoles and $\alpha^m$ magnetic dipoles orthogonal to each other, oriented along the x- and y-directions under plane-wave illumination. The two dipoles are arranged in an infinite subwavelength array with polarization rates that are Lorentzian dependent. The diameter of the nanodisk could be modified to alter the spectral location of the modes [32,33], or the geometry could be changed to cause mode mixing between the electric and magnetic dipole modes [31]. As shown in Figure 1d, coupling between the electric and magnetic dipole modes is induced by changing the symmetry-breaking of the nanodisk. The interference between the two modes produces Fano resonance with a high Q-factor. In other words, the basic theory can also be used as a reference, as described in [30]. The direction of the electric dipole moment ($p_x$) is consistent with the direction of the electric field of the incident light, and the decay of the electric dipole mode is governed by both radiative and nonradiative processes. When the symmetry of the all-dielectric nanodisk is broken, the coupling between the bright in-plane electric dipole ($p_x$) and the dark longitudinal magnetic dipole mode ($m_z$) is caused by the symmetry break. While $p_x$ dipoles are subject to both radiative and nonradiative decay processes, the $m_z$ mode only experiences nonradiative decay, and high Q-values can be achieved using low-loss dielectric materials. A similar process would occur for $p_y$ and $m_y$, as described above for $p_x$ and $m_z$.

In this work, the Fano resonance effect is used to obtain a higher Q-value. When the symmetry is broken, due to asymmetry in the structure, the center wavelength of the reflection spectrum changes. This change can be understood by considering the scattering and interference of waves.

The asymmetry in the structure leads to different parts of the optical field experiencing different phase changes, affecting the interference pattern of the waves, and thus altering the characteristics of the reflection spectrum. Specifically, the change in the center wavelength can be described by a relationship similar to the following: $\lambda_{center}$ = f (asymmetry). Here, $\lambda_{center}$ represents the change in the center wavelength, and f (asymmetry) is a function of asymmetry. The greater the asymmetry, the larger the value of this function, resulting in the change in the center wavelength in the reflection spectrum.

To further analyze the reasons for the generation of nanodisk patterns, structures of both missing wedge-shaped nanodisks ($\theta = 160°$ and $R = 170$ nm) and nanodisks under the same conditions are studied. The reflection spectra of the missing wedge-shaped nanodisk and nanodisk structures in the wavelength range of 800–1100 nm are shown in Figure 2a,b. The Fano resonance is generated by breaking the symmetry of the nanodisk, which causes the bright and dark modes to interact with each other so that the resonance peak has an extremely narrow linewidth. To explore the reasons for the formation of sharp Fano resonance spectra, the reflection modes are investigated for the two structures at different spectral positions (Figure 2c), located at wavelengths ($\lambda$) of 800 nm (M1), 950 nm (M2), 1000 nm (M3), 840 nm (M4), 938 nm (M5), and 1000 nm (M6). From the electric field diagram, the incidence of x-polarized light causes a quadrupole resonance in the

nanodisk. In the missing wedge-shaped nanodisk, dipole resonance ($\lambda$ = 938 nm) and quadrupole resonance ($\lambda$ = 840 nm) are observed. There is interference between the bright mode (quadrupole mode at $\lambda$ = 840 nm) and the dark mode (dipole mode at $\lambda$ = 938 nm), which causes the Fano resonance. The mutual interference between the dark mode and the bright mode forms an asymmetrical and sometimes sharp formant. This asymmetry is characteristic of Fano resonances, which appear as a ramp or notch in the contour of the formant, unlike traditional Lorentzian resonances. By adjusting the geometry of the metasurface, the structural parameters, or introducing special materials, the relative intensity and frequency of the dark mode and bright mode can be regulated, thus regulating the properties of the Fano resonance. The results show that, in the all-medium metasurface, by adjusting the structure of the metasurface, the mutual interference effect between dark mode and bright mode is generated; furthermore, the special spectral line profile of Fano resonance is formed under the joint action of the two modes, which expands the application direction of metasurface optics.

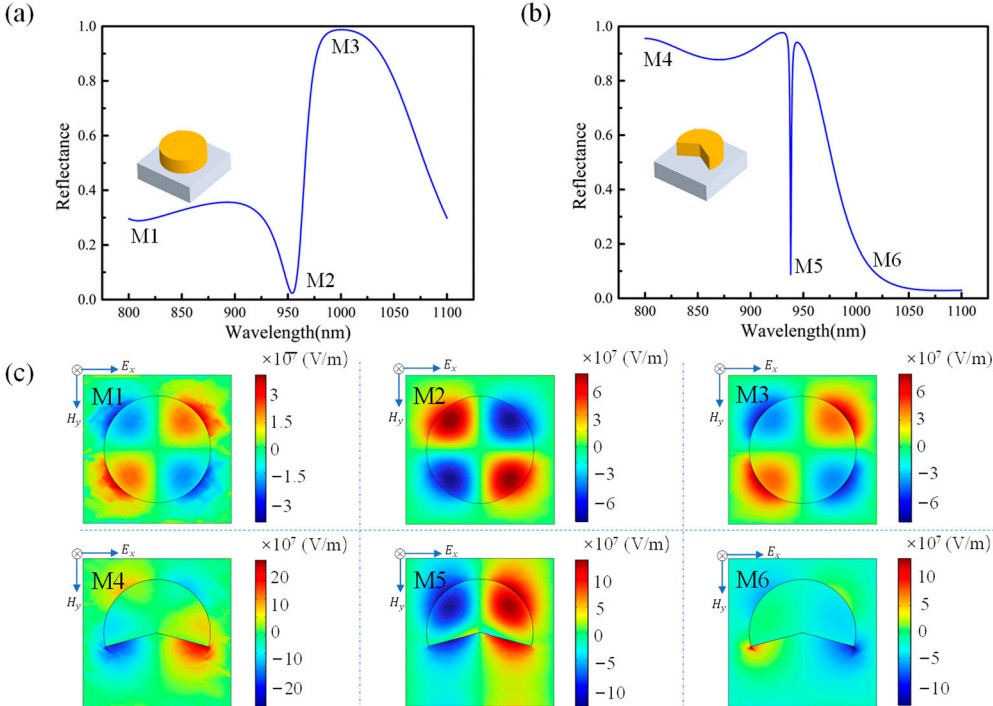

**Figure 2.** Reflection spectra and electric field distribution (*Ey*) of a metasurfaces cell. (**a**) Reflection spectra of the whole nanodisk. (**b**) Reflection spectra of the missing wedge-shaped nanodisk. (**c**) Electric field distribution (*Ey*) at different wavelength positions (800 nm (M1), 950 nm (M2), 1000 nm (M3), 800 nm (M4), 938 nm (M5), and 1000 nm (M6)).

In general, there are two main types of dipole and quadrupole modes according to the emission direction, electric vector and magnetic vector, when light is incident on the metasurface of the full medium. These modes describe the different forms of oscillations that occur when electromagnetic waves interact with matter.

Dipole modes include electric dipole mode and magnetic dipole mode. An electric dipole oscillates in one direction, usually the same or opposite direction of the electric field, and this pattern describes the oscillation of the charge distribution, similar to oscillating pairs of positive and negative charges. The direction of oscillation of a magnetic dipole is around an axis, usually the same or opposite direction of the magnetic field, and this pattern describes the oscillation of the distribution of magnetic moments, similar to a ring of current around an axis.

Quadrupole modes include electric quadrupole mode and magnetic quadrupole mode. The oscillation of an electric quadrupole involves an asymmetric oscillation of the charge

distribution, whose direction is along two orthogonal axes. This pattern describes the asymmetry of the charge distribution, similar to the oscillation of two sets of positive and negative charge pairs. The oscillation of a magnetic quadrupole involves an asymmetric oscillation of the magnetic moment distribution in a direction that is around two orthogonal axes, describing the asymmetry of the magnetic moment distribution, similar to a current loop around two orthogonal axes.

The generation and interaction of these patterns in matter is crucial for understanding and designing applications of materials such as nanostructures and metasurfaces in optics and electromagnetism. In these structures, through careful design, these modes can be regulated to achieve specific optical properties and functions.

To better design the ideal device structure, we use a simulation software based on the finite element method, which transforms the continuous physical field problem into a discrete mathematical problem. Its step size is less than 0.5 nm, and the mesh accuracy is 0.1 nm. Other boundary conditions are designed according to electromagnetic field theory.

### 3. Results and Discussion

#### 3.1. The Influence of Geometric Parameters

The effects of the missing geometrical parameters of the wedge-shaped nanodisk on the Fano resonance line shape are investigated. The missing wedge-shaped nanodisk slice angle and radius are scanned under fixed radius ($R$ = 160 nm) and slice angle ($\theta$ = 160°) conditions, respectively. Figure 3a,b show the reflection spectra of the missing wedge-shaped nanodisks for a range of slice angles and nanodisk diameters. As shown in Figure 3, the Fano resonance spectra are mainly affected by the wedges cut from the nanodisk and the radius of the nanodisk. An increase in the slice angle of the nanodisk led to a blue shift in the resonance wavelength, which led to a red shift in the resonance wavelength when the radius of the nanodisk increased. At small radii, the Fano resonance feature disappears. We are able to clearly understand the effect of the geometric parameters on the Fano resonance spectra and find the best basic unit based on the effect of the geometric parameters on the Fano resonance spectra.

The position and response level of the Fano resonance mainly depend on the contribution of the magnetic dipole and electric dipole to the Fano modes, according to $J(r) = -i\omega\varepsilon_0(n^2 - 1)E(r)$,

$$\alpha^e = \frac{1}{i\omega} \int J d^3 r \tag{2}$$

$$\alpha^m = \frac{1}{2c} \int (r \times J) d^3 r \tag{3}$$

where $n$ represents the refractive index of silicon; $r$ is the position in the Cartesian coordinate system; $\omega$ is the angular frequency; $\varepsilon_0$ is the dielectric permittivity of free space; and $J$ and $E$ represent the current and electric fields, respectively. An increase in the slice angle of the missing wedge-shaped nanodisk leads to a decrease in the edge charge density, which decreases the intensity of the Fano resonance. As the electric field distribution changed, the resonance position is blue-shifted. In addition, an increase in the radius of the missing wedge-shaped nanodisk leads to an increase in the charge density and a shift in the electric modes, which affects the position of the pattern excitation. Finally, the intensity of the Fano resonance increased, and the resonance position is red-shifted. The Fano resonance wavelength and reflection could thus be modulated by controlling the slice angle and radius of the missing wedge-shaped nanodisk, and this feature could be extended to optical filtering.

Meanwhile, if the structure is strictly symmetrical, it needs to be considered to calculate the electric and magnetic multipoles [34–36]. The graphs of these and other multipoles should be demonstrated; however, in this work, the Fano resonance caused by the asymmetry of the structure is mainly considered, and the sensor-related parameters are of most concern.

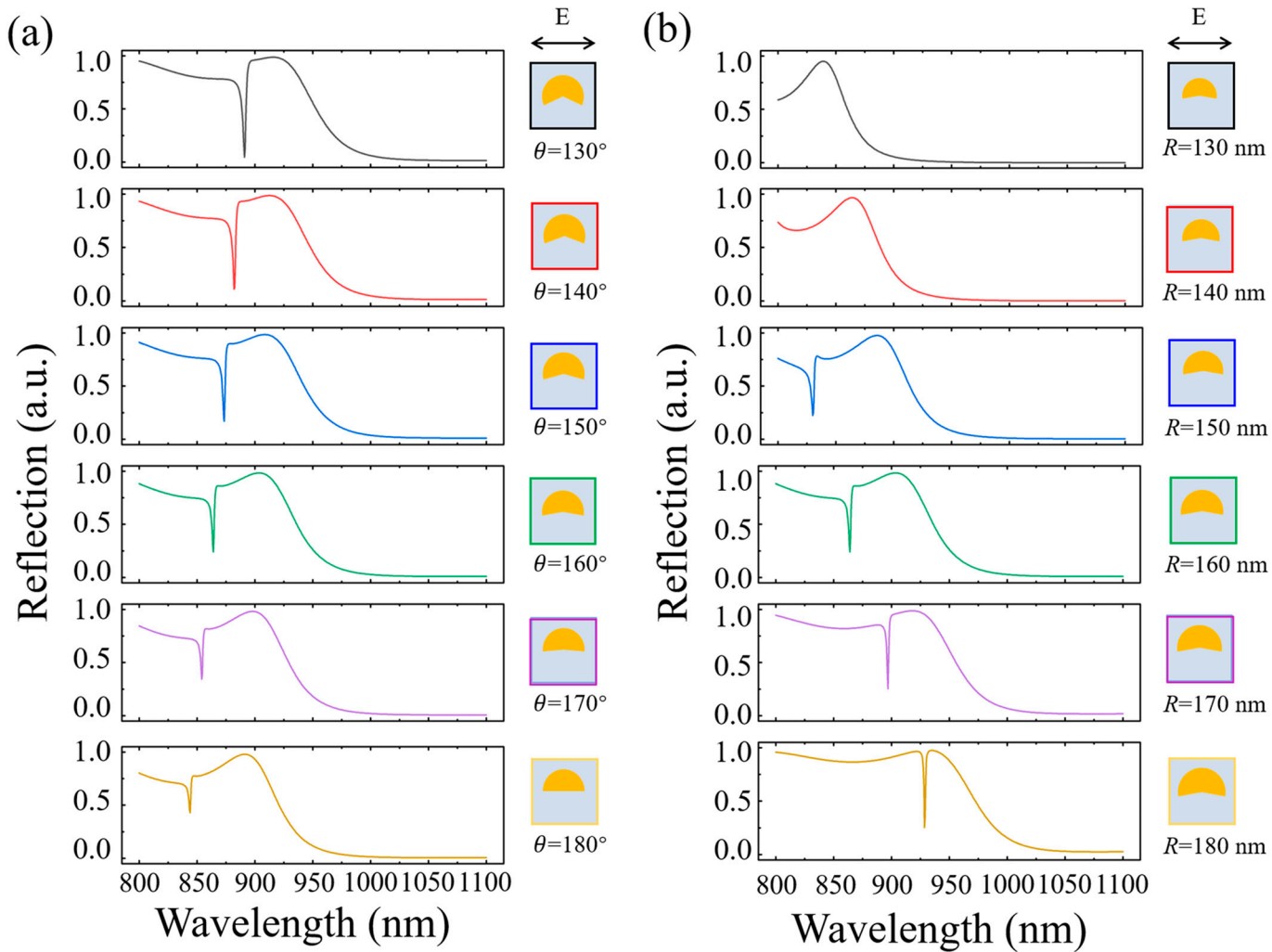

**Figure 3.** Reflection spectra at different slice angles and the radii of the nanodisk. (**a**) Reflection spectra of the missing wedge-shaped nanodisk with different $\theta$ (130–180°) at R = 160 nm. (**b**) Reflection spectra of the missing wedge-shaped nanodisk with different $R$ (130–180 nm) at $\theta$ = 160°.

To evaluate the performance of Fano resonance, the $Q$-factor of the missing wedge-shaped nanodisk is as follows [37]:

$$Q = \lambda_{Fano}/\Delta\lambda \tag{4}$$

where $\lambda_{Fano}$ is the Fano resonance wavelength and $\Delta\lambda$ is the full width at half maximum (*FWHM*) of the resonance wavelength. The maximum $Q$-factor reaches 1202 at $R$ of 180 nm and $\theta$ of 190°, while $\Delta\lambda$ is less than 1 nm. The missing wedge-shaped nanodisk had a narrow and sharp resonance peak, which could be applied for sensing.

### 3.2. Sensing Characteristics

To further understand the Fano resonance characteristics of the missing wedge-shaped silicon nanodisk, the electric field distribution of the structure is investigated. Figure 4a,c show the electric field distribution in the *x–y* and *x–z* planes of the structure, respectively, and Figure 4b,d correspond to the linear variation of the electric field distribution in the nanodisk structure. Its surface exhibits a strong electric field enhancement, which is favorable for sensing.

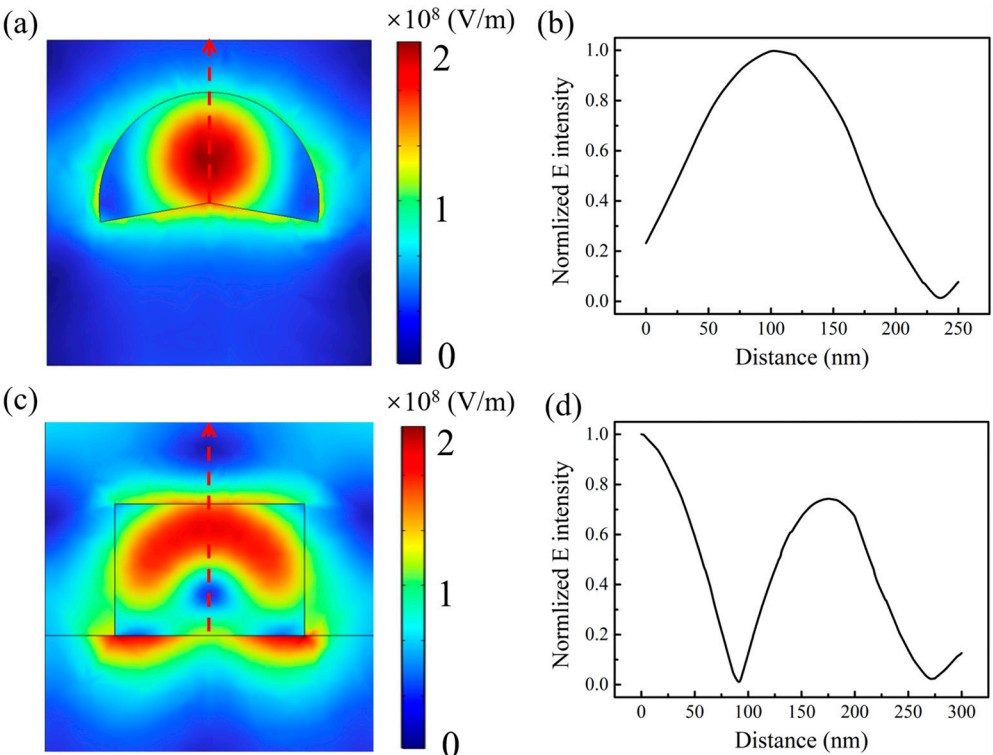

**Figure 4.** Electric field distribution of the missing wedge-shaped nanodisk ($\theta = 170°$ and $R = 180$ nm) at the resonance wavelength ($\lambda = 938$ nm). (**a**) Electric field distribution in the *x*–*y* plane. (**b**) Linear variation of the electric field distribution (normalized) in the *x*–*y* plane. (**c**) Electric field distribution in the *x*–*z* plane. (**d**) Linear variation of the electric field distribution (normalized) in the *x*–*z* plane.

The reflection spectra of the missing wedge-shaped silicon nanodisk in the ambient refractive index range of 1.33–1.39 are investigated to better visualize the sensing characteristics of the proposed structure. As shown in Figure 5a, the reflection spectra of the sensor exhibited a significant red shift with an increasing refractive index. Based on the tunable nature of the liquid environment, extremely narrow resonant peaks could be used to detect the ambient refractive index using a biosensor. With regard to the sensor performance, its sensitivity (*S*) and figure of merit (*FOM*) are important judgment indicators, where *S* is defined as follows [38]:

$$S = \Delta\lambda / \Delta n \qquad (5)$$

$\Delta\lambda$ represents the shift in the resonance wavelength and $\Delta n$ represents the change in the refractive index. The unit of $\Delta\lambda$ is nm, the unit of $\Delta n$ is the refractive index unit (RIU), and the unit of *S* is nm/RIU. The *FOM* is defined as follows:

$$FOM = S/FWHM \qquad (6)$$

The *FOM* requires both the *S* and *FWHM* to be determined. The sensitivity of the structure is approximately 83 nm/RIU. The *FOM* is approximately 92 RIU$^{-1}$. Compared to metallic materials, silicon has a sharper resonance peak and lower loss, and its resonance position could be easily observed. In the near-infrared regime, the metasurface with an all-dielectric asymmetric structure has a high *FOM*, which means that there is a very narrow formant, although the sensitivity is only 83 nm/RIU, but the formant movement is still clearly identifiable. This is not particularly advantageous in terms of sensitivity compared to metal-structured metasurface sensors, however, because the *FWHM* of the formant is small and the *Q*-value is particularly large, and even if the refractive index change is small, it can be clearly reflected in the movement of the formant. Meanwhile, there is no loss on the all-dielectric metasurface, which is also the reason for the high *Q*-value.

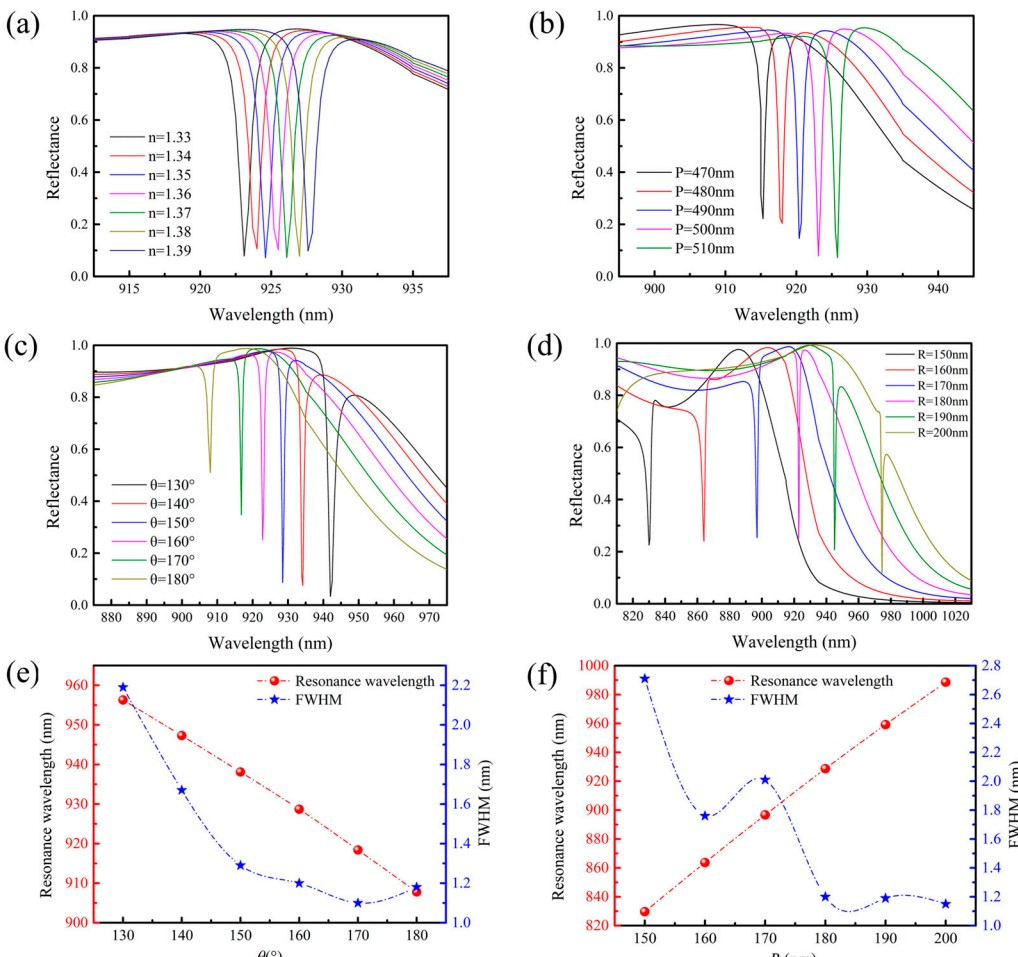

**Figure 5.** Reflection spectra and *FWHM* of the missing wedge-shaped silicon nanodisk at different parameters. (**a**) Reflection spectra changes with different refractive indices (*n* = 1.33–1.39) at *R* = 180 nm, *θ* = 170°. (**b**) Reflection spectra at different periods (*P* = 470–510 nm, at *n* = 1.33, *R* = 180 nm, *θ* = 170°). (**c**) Reflection spectra of the missing wedge-shaped nanodisk at different θ (130–180°) with *R* = 160 nm. (**d**) Reflectance spectra of the missing wedge-shaped nanodisk at different *R* (150–200 nm) with *θ* = 160°. (**e**) Evolution of the resonance wavelength and *FWHM* at *θ* of 130–180°. (**f**) Evolution of the resonance wavelength and *FWHM* at *R* of 150–200 nm and *θ* of 180°.

The effect of periodicity is investigated as shown in Figure 5b. Different periods from 470 nm to 510 nm are studied; the reflection became smaller as the array period increased, and the resonance waveform had a stable red shift. Owing to the different periods, the change in distance between the missing wedge-shaped silicon nanodisk influences the electromagnetic coupling between components. The tunability of the period could expand the performance and application of the designed metasurface based on the stability of all-dielectric materials under different conditions.

Different nanodisk radii and slice angles have sharp Fano resonances with relatively narrow linewidths and high Q-factors, and resonance linearity could be achieved by modulating the slice angle and radius of the missing wedge-shaped nanodisk. As shown in Figure 5c, the reflection spectra of the missing wedge-shaped nanodisk at different *θ* values at *R* of 160 nm show a blue shift in the resonance wavelength with an increasing slicing angle and a slight change in the line shape. The reflection spectra at different R values when *θ* is fixed at 160° are discussed (Figure 5d). Figure 5e shows the resonance wavelength and *FWHM* extracted by the Fano resonance at different slicing angles (160–190°). When the slice angle is greater than 150°, the reflectivity of the Fano resonance peak increased sharply (Figure 5f). As the radius increased, the resonance wavelength

is red-shifted, the Fano resonance feature is always present, and the overall absorption increased. Figure 5f also shows the resonance wavelength and *FWHM* extracted from the Fano resonance at different radii. The resonance wavelength varies linearly with the radius, and the *FWHM* gradually reaches saturation with an increasing radius. This is because an increase in the radius leads to an increase in the array density, which increases the silicon particle absorption. The overall change in the reflectivity of the Fano resonance peak is not significant when the radius is varied; thus, the missing wedge-shaped nanodisk array exhibits strong modulation in terms of the resonance wavelength, *FWHM*, and reflectivity. Based on the Fano resonance effect, the missing wedge-shaped nanodisk has a sharp reflection window, which is capable of reflecting up to 4% under optimal conditions and maintaining a relatively high reflectance over a wide frequency range, which makes it ideal for use as an optical filter in the near-infrared range.

Compared to disc filters and cartridge filters, which lack the flexibility to adjust and are bulky, the missing wedge-shaped nanodisk could be designed to suit the linewidth and central wavelength of a specific application. Here, reflection or transmission, *Q*-factor, and linewidth are critical factors. At *R* of 180 nm and $\theta$ of 150°, the *Q*-factor is 727 and the reflectivity is 8.7%. In addition, it maintains a relatively low transmission over a wide frequency range, making it well-suited for optical filters in the near-infrared range. Thus, the central wavelength and linewidth are designed according to the desired application. The missing wedge-shaped nanodisk structure is promising for applications, owing to the Fano effect, low-loss, and low dispersion properties of the silicon material.

### 4. Conclusions

The research has delved into a single resonator structure by disrupting the symmetry of a nanodisk to induce a Fano resonance. This design accomplishes an incredibly narrow Fano resonance, measuring less than 1 nm, within the near-infrared spectrum, boasting a remarkable *Q*-factor of 1202. Furthermore, by adjusting the radius and slice angle of the nanodisk, the resonance wavelength and linewidth can be finely modulated. The distinctively sharp and well-defined resonance peak exhibited by the wedge-shaped silicon nanodisk offers innovative avenues for biosensing, detection, and optical filtering applications. Additionally, the clarity of the obtained spectrum not only suggests the potential for expanding into other wavelength ranges and materials, but also hints at a broader spectrum of applications.

**Author Contributions:** Conceptualization, Z.G.; methodology, Y.Z.; software, Y.Z.; validation, Y.Z., Q.G. and Z.G.; formal analysis, Z.G.; investigation, Z.G.; resources, Z.G. and J.L.; data curation, Y.Z.; writing—original draft preparation, Y.Z. and Z.G.; writing—review and editing, Y.Z., Q.G., J.L. and Z.G.; visualization, Z.G.; supervision, Z.G.; project administration, Z.G. and J.L.; funding acquisition, Z.G. and J.L. All authors have read and agreed to the published version of the manuscript.

**Funding:** This research is funded by the National Natural Science Foundation of China (62075211, 62274191), the National Key Research and Development Plan of China (2023YFB3210400), and the Beijing Natural Science Foundation (4232075).

**Institutional Review Board Statement:** Not applicable.

**Informed Consent Statement:** Not applicable.

**Data Availability Statement:** The data presented in this study are available on request from the corresponding author.

**Conflicts of Interest:** The authors declare no conflicts of interest.

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
