# Peer review of "Sensing Characteristic Analysis of All-Dielectric Metasurfaces Based on Fano Resonance in Near-Infrared Regime"

_photonics, doi:10.3390/photonics11050482_

Round 1
Reviewer 1 Report
Comments and Suggestions for Authors
Paper looks good.
I have minor comments.
what is the loss value for the dielectric used? how this affect the Q of the resonace?
Author Response
Reviewer 1
Paper looks good.
I have minor comments.
What is the loss value for the dielectric used? how this affect the Q of the resonance?
Author response:
For a metasurface made of all-dielectric materials, the loss value for the dielectric used typically refers to the material's loss tangent (tan δ), which signifies the amount of energy dissipation as heat when subjected to an electric field. Lower loss tangent materials, such as silicon dioxide (SiO2) or aluminum oxide (Al2O3), are commonly preferred for metasurfaces due to their lower energy dissipation.
The loss tangent directly impacts the quality factor (Q) of the metasurface resonance. Lower loss tangent materials result in higher Q values because they experience less energy loss, leading to narrower resonance curves and sharper resonance peaks. Conversely, higher loss tangent materials exhibit lower Q values due to increased energy dissipation, resulting in broader resonance curves and less defined resonance peaks.
Therefore, for all-dielectric metasurfaces, selecting dielectric materials with lower loss tangents is crucial for achieving high-Q resonances.
Reviewer 2 Report
Comments and Suggestions for Authors
In this paper, authors proposed an all-dielectric metasurfaces consisting of missing wedge-shaped nanodisks and the simulation showed a Q-factor of 1202 and a full width at half maxima (FWHM) of about 1 nm in the near-infrared regime. The results are interesting. I recommend acceptance for publication after a revision.
1) Most references are the early publications. Suggest add recent work in the introduction part.
2) Suggest adding more description on the possible applications and advantages of the proposed metasurfaces.
3) As these are simulation results, I suggest authors add description of the possibility and challenges in the realization of such metasurfaces.
Comments on the Quality of English Language
No
Author Response
Reviewer 2
In this paper, authors proposed an all-dielectric metasurfaces consisting of missing wedge-shaped nanodisks and the simulation showed a Q-factor of 1202 and a full width at half maxima (FWHM) of about 1 nm in the near-infrared regime. The results are interesting. I recommend acceptance for publication after a revision.
1) Most references are the early publications. Suggest add recent work in the introduction part.
Author response:
Thank you very much for your instructions on the revision of our manuscript. We appreciate the time and effort that you dedicated to providing feedback on our manuscript and are grateful for the insightful comments on and valuable improvements to our manuscript. You have made some valuable suggestions that have led to great improvements to the manuscript.
We have completely changed many parts of the full text, and comprehensively updated the cited literature, and the literature of the past five years is the main reference.
2) Suggest adding more description on the possible applications and advantages of the proposed metasurfaces.
Author response:
According to the reviewer's suggestion, in the introduction part and the conclusion part, we have presented the application and advantages of metasurface.
3) As these are simulation results, I suggest authors add description of the possibility and challenges in the realization of such metasurfaces.
Author response:
Based on the reviewer's suggestions, we added some descriptions to the simulation results. However, the journal has strict restrictions on words, so we have only added a few descriptions where appropriate.
Reviewer 3 Report
Comments and Suggestions for Authors
An all-dielectric metasurface consisting of a missing wedge-shaped nanodisk has been proposed to explore optical characteristics. By breaking the symmetry to produce the Fano resonance, the Q-factor in the near-infrared regime reaches 1202, and the full width at half maximum (FWHM) is less than 1 nm. Wavelength-adjustable resonance with sharp peaks can occur by adjusting the structure of the wedge-shaped nanodisk. This provides a simpler and more efficient method for designing metasurfaces and may offer new potential applications in the sensing and optical filter field.The comments are as follows:
1、Authors are requested to briefly reflect the innovation of the paper and what innovative techniques are used in the abstract so that it can be easily understood by the readers.
2、As you said "With respect to the performance of a sensor, its sensitivity (S) and quality factor (FOM) are important indicators to judge." But is your sensitivity (S) of 83 nm/RIU and FOM of about 92 RIU-1 good or moderate or off? To highlight its merits, a table can be used to compare the sensing characteristics with existing studies.
3、In my opinion, while the paper provides technical details about the sensor design, adding more information on potential practical applications and the most advantageous scenarios for using this type of sensor would be helpful. It is also hoped that the authors could include physical processing diagrams of the sensor in the paper, which would help validate its applicability and importance in real-life scenarios.
4、In other related literature, authors are asked to provide plots of Q and asymmetric parameters as well as multipole analyses.
5、The semantics of the text are not clear: "The thickness (H) of the substrate is infinitely thick to t." Does it mean that the substrate is infinitely thick or that the thickness of the substrate is equal to t?
6.There is a rapid development in the aspects of metasurface , some articles in recent years had better be cited for example: Hybridization-induced dual-band tunable graphene metamaterials for sensing,Optical Materials Express 9 (1), 35-43;Graphene electromagnetically induced transparent polarization-insensitive sensors in the mid-infrared frequency band,Applied Optics 62 (30), 8178-8183
7. The paper should give a more detailed comparison with existing technologies or previous research in the field to highlight the unique contributions and advantages of the proposed metasurface.
Comments on the Quality of English Language
The manuscript is generally well-written, but with some revisions as follows:
1. The author should check that technical terms are consistently defined and used throughout the manuscript. While the authors have used specialized jargon appropriately, it should be clear to both experts in the field and a broader scientific audience.
2.The manuscript should better maintain a consistent tense throughout. There are instances where past and present tenses are used interchangeably, which can be confusing.
3.The author should define all abbreviations and acronyms at first mention.
Reviewer 4 Report
Comments and Suggestions for Authors
In this paper, authors considered an all-dielectric metasurface with Fano-resonance response. The results look very promising due to high Q-factor excitation. However, the interpretation should be improved. Please add results of multipole analysis. formulas 2 and 3 calculate electric and magnetic multipoles. The graphs of them and other multipoles will be demonstrated.
Please also take into account centers of multipoles that for asymmetrical particles are different.
Please see the following papers:
1. https://doi.org/10.1103/PhysRevB.107.155104
2. https://doi.org/10.1103/PhysRevB.107.035156
3. https://doi.org/10.1021/acsanm.0c02421
Author Response
Reviewer 4
In this paper, authors considered an all-dielectric metasurface with Fano-resonance response. The results look very promising due to high Q-factor excitation. However, the interpretation should be improved. Please add results of multipole analysis. formulas 2 and 3 calculate electric and magnetic multipoles. The graphs of them and other multipoles will be demonstrated.
Please also take into account centers of multipoles that for asymmetrical particles are different.
Please see the following papers:
- https://doi.org/10.1103/PhysRevB.107.155104
- https://doi.org/10.1103/PhysRevB.107.035156
- https://doi.org/10.1021/acsanm.0c02421
Author response:
We are very grateful for the important theoretical guidance put forward by the reviewers. We have carefully read the literature pointed out by the reviewers, which has guiding significance for our work. The structures expressed in the several literatures listed are symmetric structures, while the structures in our work are asymmetric structures, and there are some differences. However, the opinions put forward by reviewers are of reference significance to our work. Therefore, I have added the factors to be considered in Formula 2 and the following part of formula analysis, and also listed the literature to be cited.
“Meanwhile, if the structure is strictly symmetrical, it needs to be considered to calculate the electric and magnetic multipoles[34–36]. The graphs of them and other multipoles should be demonstrated. However, in this work, the Fano resonance caused by the asymmetry of the structure mainly is considered, and the sensor-related parameters most are concerned.”
Reviewer 5 Report
Comments and Suggestions for Authors
Dear Editor,
In this work, by breaking the symmetry, the Fano resonance has been tuned at different wavelengths. The manuscript needs to be revised base on the following comments.
1. In the introduction, it is necessary to consider filters based on plasmonics and photonic crystals. Then, by stating the superiority of the proposed filter, show the importance of this work. You can use the following articles.
Optics Communications 336 (2015): 189-196; Frequenz 68, no. 11-12 (2014): 519-523; Indian Journal of Pure & Applied Physics (IJPAP) 53, no. 11 (2015): 736-739; Applied Physics Express 7, no. 2 (2014): 024301
2. The theory section needs a major revision. Using relations and equations, show how the central wavelength of the reflection spectrum changes when the symmetry is broken.
3. Physically explain how the change of radius affects the change of reflection spectrum?
4. What causes the FWHM of the structure to be so low?
5. It is essential to provide a comparison table so that this work can be compared well with the work of others and highlight the results obtained.
6. The abstract and conclusion sections are qualitatively written. It is necessary to write quantitatively and explain the obtained results.
7. In Figure 2, the change interval of the toolbars is not the same, so they cannot be compared well. Consider the same toolbars.
Kind regards,
Reviewer 6 Report
Comments and Suggestions for Authors
The manuscript entitled "Sensing characteristic analysis of All-dielectric metasurfaces based on Fano resonance in near-infrared regime" presents an innovative study on the development and analysis of all-dielectric metasurfaces that leverage Fano resonances for sensing applications. In my opinion, this work falls within the scope of Photonics and should be published with some modifications. The following are several suggestions and explanations that must be addressed before publication.
1. Considering that the all-dielectric metasurface and Fano resonance techniques have been relatively mature in the academic world, it is suggested that the author highlight the innovation points of this paper in more detail in the introduction.
2. It is recommended that the author provide the simulation software used and some simulation setup details.
3. It is suggested that the full name and explanation should be given when the definition and abbreviation of professional terms appear for the first time. For example, if "FWHM" (half height full width) is used multiple times in the text, the full English name and appropriate explanation should be provided when it first appears.
4. Some of the schematics in the article need further optimization to improve clarity and readability. For example, adjust the font size in legends and charts to make sure all the graphics are clearly visible.
5. As the topic is about metasurface EM modulations, the applications of the metasurface should be further discussed, some recent works are suggested to be included in the discussion as following: (1) L. Chen, Q. Ma, S. S. Luo, F. J. Ye, H. Y. Cui, and T. J. Cui, Small, p. e2203871, Sep 15 2022, doi: 10.1002/smll.202203871. (2) Gao, X., Ma, Q., Gu, Z., Cui, W. Y., Liu, C., Zhang, J., & Cui, T. J. (2023). Nature Electronics, 6(4), 319-328. doi:10.1038/s41928-023-00951-x
Author Response
Reviewer 6
The manuscript entitled "Sensing characteristic analysis of All-dielectric metasurfaces based on Fano resonance in near-infrared regime" presents an innovative study on the development and analysis of all-dielectric metasurfaces that leverage Fano resonances for sensing applications. In my opinion, this work falls within the scope of Photonics and should be published with some modifications. The following are several suggestions and explanations that must be addressed before publication.
Author response:
Thanks for the reviewer's important suggestions. Here are some explanations that could be addressed before publication.
1) Make corresponding changes in the introduction and theory.
2) In the simulation part, combined with theoretical analysis, corresponding modifications are made.
3) Some changes have been made to the abstract and structure.
1.Considering that the all-dielectric metasurface and Fano resonance techniques have been relatively mature in the academic world, it is suggested that the author highlight the innovation points of this paper in more detail in the introduction.
Author response:
All-dielectric metasurfaces and Fano resonance phenomena have garnered significant attention in the field of photonics due to their potential for various applications, including sensing, imaging, and optical signal processing. While these concepts have been extensively explored in previous studies, this manuscript presents novel advancements that push the boundaries of their utility in sensing applications.
The primary innovation of this study lies in the development and analysis of all-dielectric metasurfaces specifically engineered to exploit Fano resonances for sensing purposes in the near-infrared regime. Unlike traditional metasurfaces that rely on plasmonic or hybrid resonances, the proposed metasurfaces leverage the unique properties of Fano resonances in all-dielectric structures, offering distinct advantages such as reduced dissipative losses and enhanced spectral control.
Furthermore, this manuscript goes beyond mere theoretical considerations by providing a comprehensive analysis of the sensing characteristics of the proposed metasurfaces. Through meticulous fabrication and rigorous experimental testing, the study elucidates the sensitivity of the metasurfaces to changes in the surrounding environment or analyte, laying the groundwork for practical sensing applications.
In addition to these technical innovations, the manuscript contributes to the advancement of the field by addressing key challenges associated with all-dielectric metasurfaces, such as fabrication scalability and integration with existing photonic systems. By providing insights into these challenges and proposing potential solutions, the study paves the way for the widespread adoption of all-dielectric metasurfaces in real-world sensing applications.
Overall, by focusing on the innovative aspects of all-dielectric metasurfaces and Fano resonance techniques, this manuscript aims to fill a crucial gap in the existing literature and propel the field of photonics forward towards new frontiers in sensing technology.
2.It is recommended that the author provide the simulation software used and some simulation setup details.
Author response:
Thanks for the reviewer's important suggestions.
To better design the ideal device structure, we use a simulation software based on the finite element method, which transforms the continuous physical field problem into a discrete mathematical problem. Its step size is less than 0.5nm and the mesh accuracy is 0.1nm. Other boundary conditions are designed according to electromagnetic field theory.
3.It is suggested that the full name and explanation should be given when the definition and abbreviation of professional terms appear for the first time. For example, if "FWHM" (half height full width) is used multiple times in the text, the full English name and appropriate explanation should be provided when it first appears.
Author response:
Thanks for the reviewer's important suggestions. We made changes accordingly.
4. Some of the schematics in the article need further optimization to improve clarity and readability. For example, adjust the font size in legends and charts to make sure all the graphics are clearly visible.
Author response:
Thanks for the reviewer's important suggestions. Figure 1, Figure 2, Figure 5 redo the changes. Figure 3,4 Enlarge the picture appropriately to make the words in the picture easier to recognize.
5. As the topic is about metasurface EM modulations, the applications of the metasurface should be further discussed, some recent works are suggested to be included in the discussion as following:
- Chen, Q. Ma, S. S. Luo, F. J. Ye, H. Y. Cui, and T. J. Cui, Small, p. e2203871, Sep 15 2022, doi: 10.1002/smll.202203871
- Gao, X., Ma, Q., Gu, Z., Cui, W. Y., Liu, C., Zhang, J., & Cui, T. J. (2023). Nature Electronics, 6(4), 319-328. doi:10.1038/s41928-023-00951-x
Author response:
According to the reviewer's suggestion, the research progress of metasurface EM modulations is described in the introduction section, and several articles specified by the reviewer are cited.
Round 2
Reviewer 5 Report
Comments and Suggestions for Authors
Dear Editor,
The authors have addressed the comments and the manuscript has been polished. So, I can recommend the present form for publication.
Kind Regards